# Non-Parametric Structural Priors for Geometry Theorem Prediction

**Junbo Zhao** [* 1 2 3]  **Ting Zhang** [* 1 2 3]  **Can Li** [1]  **Wei He** [4]  **Jingdong Wang** [4]  **Hua Huang** [1 2 3]

## Abstract

Multi-step theorem prediction is a central challenge in geometry problem solving. Existing neural–symbolic approaches rely heavily on supervised parametric models, which exhibit limited generalization to evolving theorem libraries. In this work, we explore training-free theorem prediction through the lens of in-context learning (ICL). We identify a critical scalability bottleneck, termed Structural Drift: as reasoning depth increases, the performance of vanilla ICL degrades sharply. We attribute this to the LLM's inability to recover latent topological dependencies, leading to unstructured exploration. To address this issue, we propose Theorem Precedence Graphs, which encode temporal dependencies from historical solution traces as directed graphs, and impose explicit topological constraints that effectively prune the search space during inference. Coupled with retrieval-augmented graph construction and a stepwise symbolic executor, our approach enables LLMs to act as structured planners without any gradient-based optimization. Experiments on the FormalGeo7k benchmark show that our method achieves 89.29% accuracy, substantially outperforming ICL baselines and matching state-of-the-art supervised models. These results indicate that explicit structural priors offer a promising direction for scaling LLM-based symbolic reasoning.

## 1. Introduction

Multi-step theorem prediction is a cornerstone of automated reasoning (Chou et al., 1993; Chou & Gao, 2001), requiring

*Equal contribution.

[1]School of Artificial Intelligence, Beijing Normal University, Beijing, China [2]Engineering Research Center of Intelligent Technology and Educational Application, Ministry of Education, Beijing, China [3]Beijing Key Laboratory of Artificial Intelligence for Education, Beijing, China [4]Baidu, Beijing, China. Correspondence to: Hua Huang <huahuang@bnu.edu.cn>.

*Proceedings of the 43rd International Conference on Machine Learning*, Seoul, South Korea. PMLR 306, 2026. Copyright 2026 by the author(s).

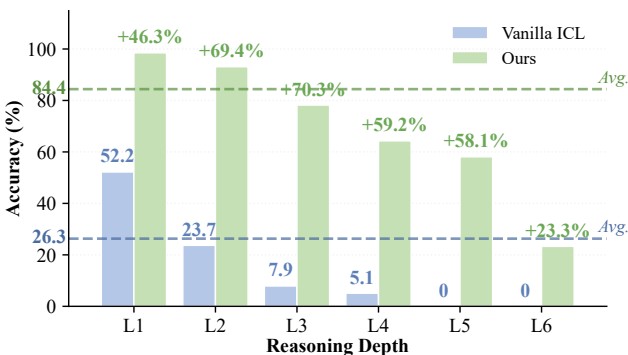

*Figure 1.* Geometry problem solving accuracy (%) across reasoning depths ($L1$–$L6$), defined by the number of invoked theorems $l$. Vanilla ICL rapidly degrades with increasing depth due to structural drift, whereas our method maintains robust performance by leveraging structural priors.

an agent to navigate complex search spaces by selecting a sequence of valid rules to achieve a goal. While this challenge spans various domains (Irving et al., 2016), it is uniquely characterized in Geometry Problem Solving (GPS) by its rigid symbolic constraints (Trinh et al., 2024; Chervonyi et al., 2025). In geometric reasoning, the applicability of each theorem is explicitly conditioned on the current set of derived facts, and any invalid step immediately breaks the reasoning chain. As the reasoning depth increases, the combinatorial explosion of potential theorem sequences makes finding a valid path exceptionally difficult (Silver et al., 2017; Lample et al., 2022).

To tackle this, modern neural–symbolic GPS pipelines typical deploy a neural model acts as a policy to propose candidate theorems (Wu et al., 2024; Peng et al., 2023), while a symbolic solver executes these steps and maintains the evolving state (Lu et al., 2021). Recent approaches predominantly rely on supervised parametric models (Zou et al., 2024; Zhang et al., 2025; Xia et al., 2024), achieving strong in-distribution accuracy but exhibiting limited flexibility. Their dependence on task-specific training over fixed theorem sets hampers generalization to unseen or expanded libraries without a costly re-training cycle.

In this work, we examine whether effective theorem prediction can be achieved without task-specific training. A natural candidate is the in-context learning (ICL) capability of LLMs (Brown et al., 2020; Wei et al., 2022). However, we observe a phenomenon termed as *structural drift*. As reason-

ing chains lengthen, the performance of vanilla ICL deteriorates sharply, in some cases approaching zero, as illustrated in Figure 1. Our analysis reveals a fundamental challenge: theorem prediction is governed by strong structural dependencies, as many theorems become applicable only after specific geometric properties have been established by earlier steps. Vanilla ICL tends to induce near-uniform action distributions over the theorem space (Baldassini et al., 2024), leading to unstructured exploration and compounding errors over long reasoning horizons (Dziri et al., 2023; Valmeekam et al., 2023; Liu et al., 2024; Berglund et al., 2023). Effective navigation of the theorem library therefore requires not only semantic understanding, but also an implicit grasp of the latent topological order underlying geometric reasoning (Kipf et al., 2019).

To this end, we propose **Pri-TPG** (**Pri**or-guided multi-step theorem prediction via **T**heorem **P**recedence **G**raphs). The core innovation of Pri-TPG is the introduction of an explicit structural guidance mechanism that augments vanilla prompts with latent topological constraints. Specifically, Pri-TPG develops *Theorem Precedence Graphs* to distill the latent temporal dependencies inherent in solution traces into a directed graph that explicitly encodes permissible orders of theorem application. Furthermore, We adopt a retrieval-augmented strategy (Lewis et al., 2020) to construct these graphs on the fly, conditioned on the input problem. In this way, Pri-TPG provides the LLM with a topological constraint that effectively prunes the combinatorial search space while requiring no gradient-based optimization.

We formulate **Pri-TPG** as a step-wise symbolic execution framework. In this architecture, the LLM functions as a planner that proposes the next theorem conditioned on the dynamic TPG, while a symbolic solver serves as the executor. Experiments on the FormalGeo7k benchmark (Zhang et al., 2024) show that our method substantially outperforms standard in-context learning baselines (Wei et al., 2022) and achieves performance comparable to supervised neural approaches (Zhang et al., 2025), while remaining entirely training-free at the theorem prediction stage.

Our contributions are summarized as follows:

- **Structural Drift:** We identify the phenomenon of *structural drift,* where ICL degrades significantly as the theorem search space grows, highlighting the need for explicit structural priors.

- **Structural Prior:** We propose Pri-TPG, a non-parametric approach that extracts query-specific structural priors from historical solution traces, providing training-free guidance for theorem prediction via Theorem Precedence Graphs.

---

Code: github.com/BNU-ERC-ITEA/Pri-TPG.

- **Empirical Performance:** Pri-TPG achieves **89.29%** accuracy on the FormalGeo7k benchmark, significantly outperforming ICL-based baselines and rivaling state-of-the-art supervised models.

## 2. Related Work

**Neural-Symbolic Geometry Problem Solving.** The evolution of GPS reflects the broader shift from expert-driven symbolic engines to large-scale neural architectures. Early systems, such as Wu's method (Wu, 1986) and the Area Method (Chou et al., 1994), offered rigorous soundness but lacked the heuristic intuition required to navigate the combinatorial explosion of auxiliary constructions. More recent work has shifted toward data-driven and neural–symbolic hybrids (Li et al., 2024; Alvin et al., 2017; Peng et al., 2023; Zou et al., 2024), including inter-GPS (Lu et al., 2021) and E-GPS (Wu et al., 2024). In parallel, large-scale formal datasets, including GeoQA (Chen et al., 2021) and FormalGeo (Zhang et al., 2024), have been introduced to support supervised learning and benchmarking of formal geometric reasoning (Zhu et al., 2024; Zhang et al., 2025). However, these approaches are highly parametric, with reasoning knowledge embedded in network weights, making adaptation to new theorem libraries costly. In contrast, our method introduces a non-parametric structural prior, enabling immediate adaptation to new theorem sets without gradient-based optimization.

**LLMs as Planners in Formal Domains.** The emergence of LLMs has shifted the focus toward agents that use formal languages (e.g., Lean, Isabelle, or domain-specific DSLs) as "tools" for reasoning (Yang et al., 2023; Huang et al., 2020; Trinh et al., 2024; Chervonyi et al., 2025). While LLMs excel at generating localized reasoning steps, they struggle with long-horizon consistency and geometric grounding, often proposing theorems that are logically valid but contextually inapplicable (Zhang et al., 2025). Current state-of-the-art methods typically rely on Reinforcement Learning (RL) to align LLM planners with symbolic verifiers (Ping et al., 2025; Xin et al., 2024). Our work diverges from this by demonstrating that the necessary "planning intuition" can be extracted directly from historical solution traces via dynamic theorem precedence graphs, bypassing the need for RL-based alignment.

**Retrieval-Augmented Reasoning.** Retrieval-Augmented Generation (RAG) is traditionally used to provide LLMs with factual context to mitigate hallucinations (Lewis et al., 2020; Guu et al., 2020). While early methods focused on static document retrieval, recent extensions incorporate tool-mediated outputs to support complex reasoning tasks (Paranjape et al., 2023). To better capture interdependent concepts, GraphRAG frameworks (Edge et al., 2024; Sarthi et al., 2024; Jimenez Gutierrez et al., 2024; Kabir et al., 2025)

have transitioned from flat vector search to relational retrieval via pre-constructed knowledge graphs. However, these methods primarily structure query intent and still inject retrieved information as unstructured prompt context. In contrast, we target the downstream reasoning process by imposing structure on the retrieved content itself. By formalizing retrieved solution traces into a directed precedence graph, we introduce an explicit structural prior that constrains the LLM's action space, marking a shift from content-augmented RAG to structure-augmented reasoning.

## 3. Methodology

In this section, we formalize multi-step geometry theorem prediction as a constrained symbolic planning problem and present a training-free neural-symbolic framework. Our approach introduces a *precedence-induced structural prior* over theorem usage, instantiated as a Theorem Precedence Graph, and combines it with multimodal retrieval and state-aware symbolic execution to guide LLMs toward efficient and valid problem solving.

### 3.1. Problem Formulation

A geometry problem is defined as a tuple $\mathcal{P} = (\mathcal{T}, \mathcal{D}, \mathcal{S}_0, g)$, where $\mathcal{T}$ denotes the textual description, $\mathcal{D}$ the diagram, $\mathcal{S}_0$ the initial symbolic state representing a set of predicates, and $g$ the target goal. Consistent with prior work (Zhang et al., 2025), we focus on the theorem prediction task, assuming the initial formal state $\mathcal{S}_0$ is provided.

Let $\mathcal{L}$ be a finite theorem library consisting of hundreds of theorems and $\text{Solver}(\cdot)$ a symbolic executor. Multi-step theorem prediction seeks a sequence of theorem applications $\mathcal{A} = \{a_1, \ldots, a_T\}, a_t \in \mathcal{L}$, that induces a valid state transition sequence,

$$\mathcal{S}_0 \xrightarrow{a_1} \mathcal{S}_1 \xrightarrow{a_2} \cdots \xrightarrow{a_T} \mathcal{S}_T, \quad \text{such that } g \subseteq \mathcal{S}_T.$$

Each transition is governed by symbolic preconditions,

$$\mathcal{S}_t = \begin{cases} \text{Solver}(\mathcal{S}_{t-1}, a_t), & \text{if } \text{Pre}(a_t) \subseteq \mathcal{S}_{t-1}, \\ \bot, & \text{otherwise}, \end{cases}$$

where $\text{Pre}(a_t)$ denotes the logical prerequisites of theorem $a_t$ and $\bot$ represents an invalid transition.

This formulation highlights the core challenge: while the symbolic executor enforces correctness, the space of theorem sequences grows combinatorially with proof length, making prediction from flat theorem distribution ineffective.

### 3.2. Precedence-Induced Structural Prior

Geometric proofs exhibit strong causal dependencies: certain theorems must precede others to establish necessary constructions or relations. To encode this inductive bias, we introduce the **Theorem Precedence Graph**, a directed graph $G = (V, E)$, where each node $v \in V \subseteq \mathcal{L}$ represents a theorem. A directed edge $(u \to v) \in E$ exists if the conclusion of theorem $u$ provides a necessary prerequisite for applying theorem $v$ in a problem.

Instead of casting theorem selection as an unstructured classification problem, the TPG imposes a topological prior that converts search into a guided traversal over admissible partial orders. For each solved instance, the TPG is induced from the solution trace by extracting symbolic dependencies among theorem applications. Rather than constraining the model to fixed paths, the TPG serves as a global structural reference, aligning local theorem predictions with the overarching logic of geometric deduction.

**Search Space Factorization.** In a straightforward search setting, the action space at each reasoning step spans the entire library $|\mathcal{L}|$, leading to an intractable combinatorial explosion $|\mathcal{L}|^H$ where $H$ is the number of reasoning steps. We introduce a structure-dependent prior derived from the TPG to prune the search volume. Crucially, instead of applying this prior as a static global constraint, we achieve precise guidance by transforming this structure into a query-adaptive prior that captures problem-contextual dependencies, and a state-aware prior that provides granular guidance at each prediction step. In the following sections, we formally detail the formulation of query-adaptive and state-aware priors.

### 3.3. Query-Adaptive Prior via Multimodal Retrieval

Recognizing that theorem precedence is inherently context-sensitive, we propose to induce a query-specific TPG by leveraging historical reasoning patterns from semantically similar problems. Given a target query $P_q$, we employ a multimodal encoder $\phi(\cdot)$ (Günther et al., 2025) to map its textual description, visual diagrams, and symbolic initial states into a joint embedding space. To identify relevant structural priors, we perform a nearest-neighbor search over a problem database $\mathcal{B} = \{P_i\}_{i=1}^N$, retrieving the top-$K$ most analogous instances,

$$\mathcal{N}_K(P_q) = \arg \text{ top-}K_{P_i \in \mathcal{B}} \, \text{sim}(\phi(P_q), \phi(P_i)). \quad (1)$$

The retrieved set provides a curated candidate space,

$$\mathcal{L}_q = \bigcup_{P_i \in \mathcal{N}_K(P_q)} \text{Theorems}(P_i), \quad (2)$$

where $\text{Theorems}(P_i)$ denotes the set of theorems used to solve $P_i$. Further, we synthesize the local structures into a unified query-specific graph,

$$G_q = \bigcup_{P_i \in \mathcal{N}_K(P_q)} \text{TPG}(P_i), \quad (3)$$

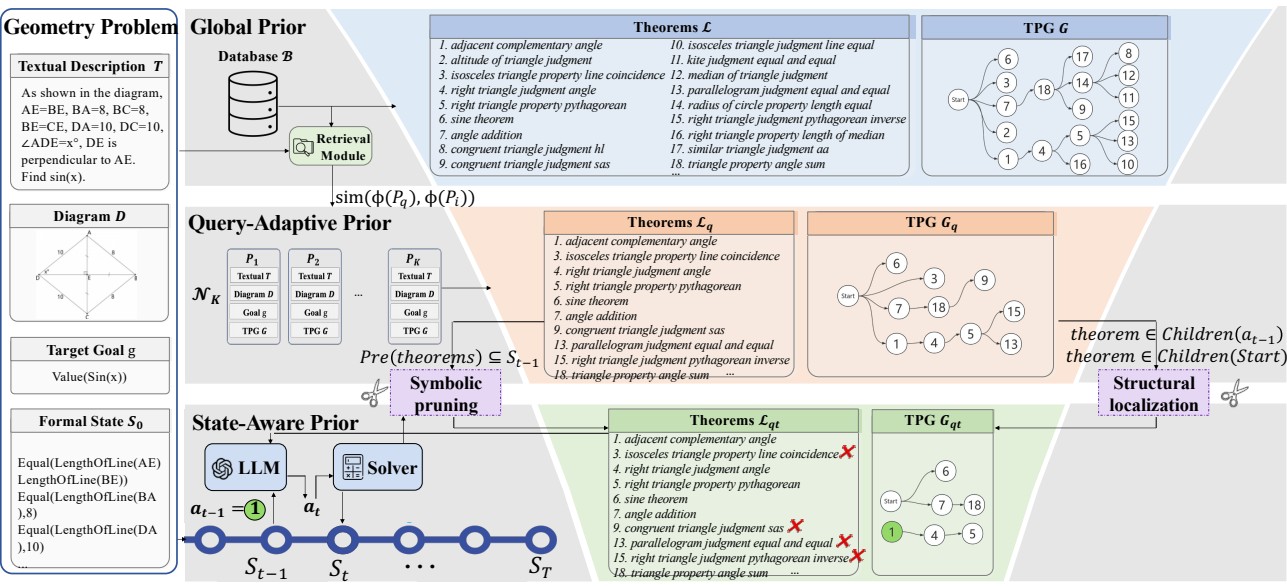

*Figure 2.* Overview of our Pri-TPG workflow, where we successively refine the structural prior to provide precise guidance.

resulting in $G_q = (V, E)$, where $V \subseteq \mathcal{L}_q$, and each directed edge $(u \to v)$ is assigned a normalized weight $w(u \to v)$ according to its frequency,

$$w(u \to v) = \frac{1}{K} \sum_{i=1}^{K} \mathbb{I}\big[(u \to v) \in E_i\big]. \quad (4)$$

Finally, we augment $G_q$ with a virtual START node, which encodes the prior probabilities for initiating theorem sequences, thereby providing a grounded entry point for the subsequent state-aware reasoning.

### 3.4. State-Aware Prior via Symbolic Validation

Instead of generating an entire theorem sequence in a single pass, an approach susceptible to error accumulation, we formulate theorem selection as an interleaved, iterative decision process. By tightly coupling the LLM with the symbolic solver, the framework exploits real-time execution feedback to construct a state-aware prior that progressively refines the search space. Specifically, at each reasoning step $t$, we apply two complementary *filtering mechanisms*,

- **Symbolic Pruning of $\mathcal{L}_q$ to $\mathcal{L}_{qt}$:** Each candidate theorem $v \in \mathcal{L}_q$ is verified by the symbolic solver against the current symbolic state $\mathcal{S}_t$. Only theorems whose prerequisites are type-consistent are retained, thereby eliminating mathematically invalid actions.

- **Structural Localization of $G_q$ to $G_{qt}$.** Let $a_{t-1}$ be the theorem applied at the previous step (or the START node when $t = 1$). We only retain the descendants of $a_{t-1}$ / START in the query-specific graph $G_q$.

**Candidate Prioritization.** To further steer the LLM's decision process, the remaining valid candidates are prioritized using a composite scoring function,

$$\Psi(v) = \alpha s_{\text{goal}}(v) + \beta s_{\text{graph}}(v) - \gamma s_{\text{hist}}(v), \quad (5)$$

where (i) $s_{\text{goal}}$ measures the semantic similarity between theorem $v$'s expected conclusion and the target goal $g$, encouraging backward-oriented reasoning, (ii) $s_{\text{graph}}$ exploits the edge weights in $G_q$ to explicitly promote the immediate successors of the previous action $a_{t-1}$, including both first-order and second-order child nodes, by assigning them additional scores proportional to their corresponding edge weights, thereby biasing the selection toward locally coherent and empirically supported theorem transitions, and (iii) $s_{\text{hist}}$ penalizes theorems that have been invoked, serving as a critical anti-looping mechanism to prevent redundant reasoning trajectories.

### 3.5. Structural Prior-Augmented LLM Prediction

We formulate each reasoning step as a constrained generation task. The LLM prompt is augmented with the top-ranked candidate set $\mathcal{L}_{qt}$, the pruned TPG $G_{qt}$, and the execution history from the previous five steps. This structured context enables the LLM to act as a high-level reasoner, selecting the most promising action $a_t$ from a verified and prioritized subset, thereby bridging flexible neural generation and rigorous symbolic deduction. Figure 2 presents the overall pipeline, with additional implementation details and prompt templates provided in Appendix.

### 3.6. Discussion

**Structural Induction over Parametric Heuristics.** State-of-the-art solvers such as AlphaGeometry (Trinh et al.,

2024) rely on parametric strategy functions trained over millions of synthetic proofs. In contrast, our framework adopts a non-parametric approach grounded in structural induction. While still leveraging a corpus of historical solutions, it entirely avoids gradient-based optimization. Our central insight is that theorem applications exhibit strong local structural regularities, which we explicitly capture through a precedence graph formulation.

**Space Contraction over Unconstrained Search.** The proposed framework fundamentally reconfigures the search dynamics of symbolic solvers. Supplying the LLM with a pruned and ranked candidate set reduces the per-step selection complexity from $O(|\mathcal{L}|)$ to $O(|\mathcal{L}_{qt}|)$, where $|\mathcal{L}_{qt}| \ll |\mathcal{L}|$. Taking our experiments on FormalGeo7K as an example, where we take $|\mathcal{L}_{qt}| \approx 30$, compared to $|\mathcal{L}| = 300$, the instantaneous search space is pruned by 90% in a single reasoning step. This contraction compounds over longer derivations, effectively mitigating the "search-depth bottleneck" that typically plagues long-chain symbolic reasoning.

## 4. Experiments

### 4.1. Settings

**Datasets.** We evaluate on three formal geometry benchmarks: **FormalGeo7K** (Zhang et al., 2024), **Geometry3K** (Lu et al., 2021), and **GeoQA** (Chen et al., 2021). Geometry3K contains three thousand diagram-text geometry problems with formal-language parsing annotations and a standard train/validation/test split; it covers diverse geometric primitives (e.g., lines, triangles, circles, quadrilaterals). GeoQA provides near five thousand geometry problems with annotated programs that describe explicit solving procedures, serving as a testbed for structured numerical reasoning. FormalGeo7K is a larger-scale formal geometry dataset with natural language descriptions, geometric diagrams, ground-truth formal language annotations, and theorem-sequence supervision, designed to support scalable theorem prediction under a consistent formal system. Our main experiments are on FormalGeo7K using its standard test split of 1,400 problems. Following Zhang et al. (2025), we stratify test problems into six difficulty levels (L1–L6): L1 ($l \leq 2$), L2 ($3 \leq l \leq 4$), L3 ($5 \leq l \leq 6$), L4 ($7 \leq l \leq 8$), L5 ($9 \leq l \leq 10$), and L6 ($l \geq 11$), where $l$ is the number of theorems invoked in the problem.

**Evaluation.** We report *accuracy* (%) as the proportion of problems solved within a fixed per-problem timeout of 600 seconds. A problem is considered solved if the solver's final formal conclusion exactly matches the ground-truth target under symbolic verification. To isolate the reasoning and search capabilities, we intentionally exclude the evaluation of upstream parsing and formalization components. Accordingly, all solvers are provided with ground-truth formal

inputs, including the textual problem specification and the corresponding diagram representation. This evaluation protocol is applied uniformly across all baselines to ensure a fair and controlled comparison. To ensure a rigorous evaluation, we guarantee that there is no retrieval leakage in our pipeline. The retrieval database is constructed exclusively from the training split of the dataset, and all test-time retrieval queries adhere to strict hold-out separation. The details are documented in Appendix A.

**Baselines.** We compare our method against representative baselines. **LLM-only baselines** solve geometry problems in an end-to-end fashion using LLMs. These approaches generate answers solely through natural language reasoning, without invoking a symbolic solver, and thus rely entirely on the model's implicit reasoning capability. **Neural-symbolic solvers** integrate neural models with a symbolic reasoning engine to perform theorem-driven search. We further divide them into *training-based* and *training-free* variants. *Training-based neural-symbolic solvers* fine-tune a pre-trained neural network to learn a policy for theorem selection, which is then combined with graph- or tree-based symbolic search to heuristically explore the solution space. *Training-free neural-symbolic solvers* leverage pre-trained LLMs for theorem prediction and interface them with a symbolic solver at inference time, avoiding any task-specific training. Our direct baseline, *Vanilla ICL*, retains the stepwise executor interaction protocol but removes the structural prior, requiring the planner to select candidate theorems from the full library at each step.

### 4.2. Results on FormalGeo7K

Table 1 presents the performance of our proposed framework compared to various baselines on the FormalGeo7K dataset (Zhang et al., 2024). Our proposed method, particularly Pri-TPG (GPT-5.2), achieves state-of-the-art performance with an overall accuracy of 89.29%. This significantly outperforms the strongest LLM-only baseline, Claude 4.5 Sonnet (75.79%), and surpasses the best training-based neural-symbolic solver, FGeo-HyperGNet (88.36%). Notably, our framework achieves these results in a training-free manner, demonstrating the superior generalization ability of our state-aware prior guidance.

**Comparison with LLM-only Direct Solving.** The results highlight a substantial "reasoning gap" in the generative capabilities of LLMs. While GPT-5.2 achieves 73.14% through direct solving, our framework boosts its performance to 89.29% (+16.15%). The improvement is even more pronounced for smaller models. Pri-TPG (GPT-5 mini) reaches 84.42%, which is nearly 20% higher than its direct-solving counterpart (64.79%). This suggests that without external symbolic scaffolding, LLMs are prone to error accumulation, leading to invalid reasoning paths.

*Table 1.* Comparison results between different methods on FormalGeo7K (*accuracy*, %). All methods are evaluated on the standard test set (1400 problems) under the same per-problem timeout (600s) and take ground-truth parsed formal inputs as input. Our proposed method, Pri-TPG (GPT-5.2), achieves state-of-the-art performance, significantly outperforms the strongest LLM-only baseline, and even surpasses the best training-based neural-symbolic solver.

| Method | Total (1400) | L1 (479) | L2 (376) | L3 (266) | L4 (157) | L5 (62) | L6 (60) |
|---|---|---|---|---|---|---|---|
| **LLM-only (direct solving)** | | | | | | | |
| DeepSeek v3.2 (Liu et al., 2025) | 57.79 | 69.94 | 52.93 | 58.65 | 49.04 | 37.10 | 31.67 |
| GPT-5 mini (OpenAI, 2025a) | 64.79 | 74.11 | 63.30 | 64.66 | 53.50 | 53.23 | 41.46 |
| GPT-5.2 (OpenAI, 2025b) | 73.14 | 80.38 | 73.40 | 74.81 | 63.06 | 59.68 | 46.67 |
| Claude 4.5 Sonnet (Anthropic, 2025) | 75.79 | 84.55 | 73.94 | 76.32 | 67.52 | 64.52 | 48.33 |
| Qwen3-VL (Bai et al., 2025) | 65.93 | 74.53 | 65.43 | 72.18 | 50.96 | 41.94 | 36.67 |
| Doubao seed 1.8 (ByteDance, 2025) | 69.14 | 74.11 | 69.15 | 71.43 | 64.33 | 50.00 | 51.67 |
| **Neural-symbolic solvers (training-based)** | | | | | | | |
| Inter-GPS (Lu et al., 2021) | 60.50 | 76.20 | 63.30 | 60.90 | 39.49 | 17.74 | 15.00 |
| NGS (Chen et al., 2021) | 62.60 | 62.22 | 64.97 | 72.79 | 57.47 | 56.41 | 36.59 |
| DualGeoSolver (Xiao et al., 2024) | 62.11 | 62.96 | 67.80 | 65.44 | 60.92 | 53.85 | 34.15 |
| FGeo-TP (He et al., 2024) | 80.86 | 96.43 | 85.44 | 76.12 | 62.26 | 48.88 | 29.55 |
| FGeo-DRL (Zou et al., 2024) | 80.85 | 97.61 | 91.88 | 70.82 | 57.55 | 36.17 | 27.59 |
| FGeo-HyperGNet (Zhang et al., 2025) | 88.36 | 96.24 | 91.76 | 87.59 | **82.17** | 56.45 | **56.67** |
| **Neural-symbolic solvers (training-free)** | | | | | | | |
| ForwardSearch (Zhang et al., 2024) | 39.71 | 58.47 | 41.01 | 34.16 | 16.40 | 5.45 | 4.79 |
| BackwardSearch (Zhang et al., 2024) | 35.44 | 66.43 | 34.98 | 11.78 | 6.56 | 6.09 | 1.03 |
| Vanilla ICL (GPT-5 mini) | 26.29 | 52.19 | 23.67 | 7.89 | 5.10 | 0.00 | 0.00 |
| **Pri-TPG (GPT-5 mini)** | 84.42 | 98.54 | 93.09 | 78.20 | 64.33 | 58.06 | 23.33 |
| **Pri-TPG (GPT-5.2)** | **89.29** | **99.16** | **96.28** | **87.92** | 77.07 | **66.13** | 30.00 |

*Table 2.* Comparison results between different methods on Geometry3K and GeoQA (*accuracy*, %). Our method achieves the best reported accuracy on Geometry3K, while remaining highly competitive on GeoQA. – indicates results not reported due to closed-source implementations or structural parsing incompatibilities. * denotes evaluation on the established 92%+ subset aligned with HyperGNet to remove duplicate-diagram problems and ensure a fair comparison.

| Method | Geometry3K | GeoQA |
|---|---|---|
| GeoDRL (Peng et al., 2023) | 85.87 | – |
| SCA-GPS (Ning et al., 2023) | – | 64.10 |
| DualGeoSolver (Xiao et al., 2024) | – | 65.20 |
| FGeo-HyperGNet* (Zhang et al., 2025) | 91.99 | **85.64** |
| Pri-TPG (GPT-5.2)* | **95.16** | 85.02 |

**Comparison with Training-based Solvers.** Our training-free framework demonstrates superior zero-shot generalization and robustness in mid-range complexity tasks. Specifically, Pri-TPG (GPT-5.2) achieves near-perfect performance in levels L1–L3 (e.g., 99.16% at L1), consistently outperforming fine-tuned baselines by leveraging the broad reasoning priors of LLMs constrained by the structural prior. Although a performance gap emerges in the highest difficulty tiers (L4–L6), our approach offers a more scalable and resource-efficient alternative that bypasses the intensive data dependency and computational overhead of task-specific training. More importantly, this retrieval-based

prior guidance is inherently orthogonal to parametric training pipelines; it can function as a plug-and-play search module that integrates seamlessly into existing trained neuro-symbolic solvers, offering a promising avenue to further elevate their performance ceilings.

**Performance of Difficulty Levels (L1-L6).** As problem complexity increases, all methods exhibit a consistent performance degradation. Our approach remains robust through L5, where Pri-TPG (GPT-5.2) attains 66.13%, surpassing the strongest training-based baseline by nearly 10%. At the most challenging level (L6), our performance declines relative to FGeo-HyperGNet, indicating that specialized architectures may retain advantages in extremely long-horizon reasoning. Notably, the collapse of Vanilla ICL at L5-L6 demonstrates that exemplar-based prompting alone is insufficient for complex formal proofs, underscoring the necessity of the TPG's structured guidance to effectively constrain the exponentially expanding search space.

### 4.3. Results on Geometry3K and GeoQA

Table 2 reports comparison with representative *training-based* neural-symbolic geometry solvers on Geometry3K (Lu et al., 2021) and GeoQA (Chen et al., 2021). In particular, our method achieves the best reported accuracy on Geometry3K, while remaining highly competitive on GeoQA. This performance gap is notable given that

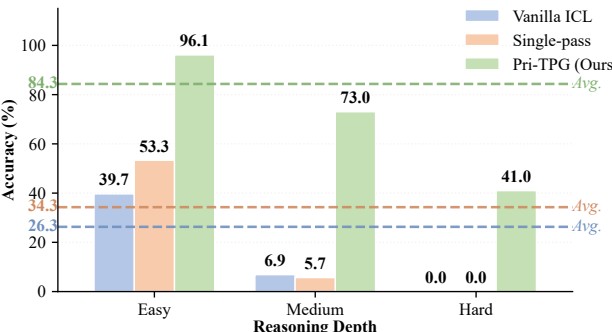

*Figure 3.* Illustrating the effect of symboic feedback. Single-pass predicts all theorems in one pass with query-adaptive prior. While it outperforms Vanilla ICL, it substantially underperforms Pri-TPG, demonstrating the importance of iterative symbolic feedback.

most competing methods rely on dataset-specific training, whereas our framework operates in a training-free manner.

## 4.4. Analysis

We conduct comprehensive ablation studies to systematically analyze the contribution of each component in our framework. To balance clarity and completeness while avoiding wide tables, we report macro-averaged accuracy on **Easy** (L1–L2), **Medium** (L3–L4), and **Hard** (L5–L6). For cost-efficiency, all ablations are performed using GPT-5 mini within our framework.

**Symbolic Feedback is Essential for Logical Error Correction.** To evaluate the role of symbolic feedback, we compare our framework with a *single-pass* variant in which the LLM generates the entire theorem sequence in a single forward pass, while still conditioned on the query-adaptive TPG, and the symbolic executor verifies it only post hoc. This design eliminates intermediate feedback and thus precludes mid-course correction. As shown in Figure 3, removing iterative interaction causes a catastrophic performance drop: overall accuracy falls to 34.3% and collapses to 0 on high-difficulty problems. These results demonstrate that formal reasoning is inherently a closed-loop process, in which real-time symbolic feedback is indispensable for successful theorem proving.

**RAG and TPG are Effective Non-parametric Structural Priors.** We evaluate the impact of non-parametric structural priors by ablating key components of our framework. The results are shown in Table 3. In *Vanilla ICL*, the iterative symbolic feedback is retained, but both RAG and TPG are removed, forcing the planner to select from the entire theorem library at each step. Accuracy under this setting drops sharply to 26.29% overall and 0% on Hard problems, indicating that iterative verification alone cannot overcome the combinatorial complexity of a large, weakly

---

Macro-average gives equal weight to each difficulty level.

---

*Table 3.* Illustrating the effect of non-parametric structural priors on multi-step theorem reasoning. Accuracy drops sharply when both RAG and TPG are removed; introducing RAG and TPG progressively improves performance, highlighting the importance of candidate narrowing and explicit precedence priors.

| Setting | Iterative | RAG | TPG | Total | Hard |
|---|---|---|---|---|---|
| Vanilla ICL | ✓ | ✗ | ✗ | 26.29 | 0.00 |
| w/o TPG | ✓ | ✓ | ✗ | 72.64 | 22.95 |
| Pri-TPG | ✓ | ✓ | ✓ | **84.42** | **40.98** |

*Table 4.* Illustrating the effect of increasingly precise structural priors, showing that each successive refinement consistently improves performance.

| Setting | Total | Easy | Medium | Hard |
|---|---|---|---|---|
| Vanilla ICL | 26.29 | 39.65 | 6.86 | 0.00 |
| Global prior | 29.71 | 40.70 | 14.89 | 4.10 |
| Query-adaptive prior | 58.43 | 72.40 | 41.61 | 18.85 |
| Pri-TPG (state-aware prior) | **84.42** | **96.14** | **73.05** | **40.98** |

structured action space. Introducing **RAG**, which provides solver-validated candidate theorems at each step, markedly improves performance to 72.64%, highlighting the importance of narrowing the action space using RAG. Incorporating our **TPG** further guides theorem selection, raising accuracy to 84.42%. This result demonstrates that candidate narrowing alone is insufficient: explicit precedence priors encoded by TPG are essential for directing reasoning trajectories, preventing unproductive detours, and sustaining efficiency as proof length increases.

**Increasingly Precise Structural Priors Systematically Improve Multi-Step Reasoning.** We analyze how progressively refined structural priors enhance multi-step theorem reasoning, with quantitative results summarized in Table 4. Starting from *Vanilla ICL*, which lacks explicit prior guidance, we first introduce a **global prior** that encodes all theorem precedence patterns, providing overall structural regularization and yielding a clear performance gain. We then incorporate a **query-adaptive prior** via multimodal retrieval over semantically similar problems, which further improves accuracy by conditioning the prior on problem-specific semantics. Finally, we introduce a **state-aware prior** through tight integration with symbolic validation, enabling step-wise refinement aligned with both the query context and the evolving reasoning state. As shown in Table 4, each successive refinement consistently improves performance, demonstrating that increasingly precise priors effectively constrain the reasoning trajectory, reduce unproductive exploration, and guide the model toward valid proofs. Overall, these results highlight that, in reasoning systems with large action spaces, precise prior guidance is essential to harness the model's full potential.

*Table 5.* Illustrating the effect of different LLMs, showing robust performance and highlighting our method's plug-and-play nature. Complete results in Appendix B.1

| Backbone | Easy | Medium | Hard |
|---|---|---|---|
| DeepSeek v3.2(Liu et al., 2025) | 95.56 | 72.10 | 39.34 |
| GPT-5 mini(OpenAI, 2025a) | 96.14 | 73.00 | 40.98 |
| Claude 4.5 Sonnet(Anthropic, 2025) | 97.31 | 77.54 | 48.36 |
| Gemini 3.0 Pro(DeepMind, 2025) | 97.54 | 81.56 | 48.36 |
| GPT-5.2(OpenAI, 2025b) | **97.89** | **83.92** | **48.36** |

**Robust Performance Gains Across Different LLMs.** To demonstrate that our framework serves as a general-purpose reasoning scaffold rather than a model-specific optimization, we evaluate it across a diverse set of LLM backbones under identical agent configurations. As shown in Table 5, the resulting performance gains are highly consistent across models. Although absolute accuracy scales with the intrinsic capacity of the base model, as expected for LLM-driven planners, the framework consistently achieves strong success rates. These results demonstrate the robustness and scalability of the proposed approach, highlighting its plug-and-play nature and its ability to seamlessly benefit from future advances in LLM capabilities without retraining.

*Table 6.* Illustrating the effect of retrieval pool size $K$, showing that accuracy increases monotonically with larger $K$.

| Top-$K$ | Total | Easy | Medium | Hard |
|---|---|---|---|---|
| 15 | 71.86 | 83.37 | 52.96 | 28.69 |
| 30 | 79.00 | 94.39 | 61.64 | 30.33 |
| 100 | 80.29 | 95.32 | 64.30 | 30.33 |
| 200 | **84.42** | **96.14** | **73.05** | **40.98** |

**The Effect of Retrieval Pool Size $K$.** We examine the effect of retrieval pool size by varying the number of retrieved neighbors $K$ while holding all other components fixed. (1) As shown in Table 6, overall accuracy increases monotonically with larger $K$, with the largest gains observed on the Medium and Hard subsets. This pattern reflects a clear *recall effect*: harder problems typically require longer and more specialized reasoning chains. When $K$ is small, critical theorems are often omitted from the candidate set, creating a hard bottleneck that prevents proof completion regardless of the LLM's reasoning capability. Increasing $K$ alleviates this issue by improving the likelihood that the necessary theorems are retrieved. (2) We further analyze retrieval effectiveness using coverage-based metrics that are independent of the executor. Specifically, we report two complementary measures as functions of top-$K$ retrieval: (i) **problem coverage**, defined as the fraction of test problems for which *all* ground-truth required theorems are contained

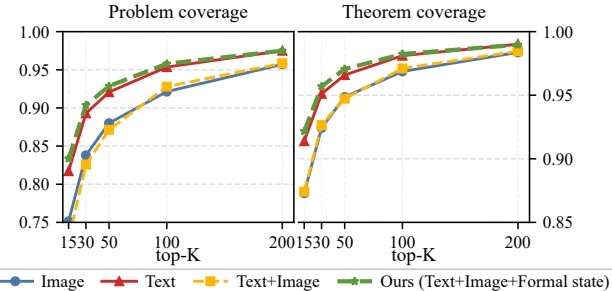

*Figure 4.* Illustrating the problem coverage and theorem coverage according to different $K$ under various query conditions.

in the retrieved candidate set,

$$\mathcal{C}_{prob} = \frac{1}{M} \sum_{i=1}^{M} \mathbb{I}(\mathcal{T}_i^* \subseteq \mathcal{L}_{P_i}), \qquad (6)$$

and (ii) **theorem coverage**, defined as the fraction of distinct ground-truth theorems that appear in the retrieved sets across the entire test corpus,

$$\mathcal{C}_{th} = \frac{\sum_{i=1}^{M} |\mathcal{T}_i^* \cap \mathcal{L}_{P_i}|}{\sum_{i=1}^{M} |\mathcal{T}_i^*|}, \qquad (7)$$

where $\mathcal{T}_i^*$ is the set of ground-truth theorems used to solve $P_i$, $M$ is the total number of test problems. Figure 4 shows that both coverage metrics increase steadily with $K$ and begin to saturate around $K \approx 100$–$200$, closely mirroring the accuracy trends observed in Table 6.

**The Effect of Retrieval Query Conditions.** We compare different retrieval query conditions in Figure 4. Notably, naive multi-modal fusion (text+image) does not yield additive gains, likely due to modality gaps in representation. Our approach grounds both modalities in a unified formal language, effectively bridging this gap and achieving the highest coverage. However, coverage alone is insufficient: even when all relevant theorems are retrieved, the planner must still correctly select and order them across multiple steps. Our TPG guidance introduces an explicit theorem-precedence prior that complements retrieval, stabilizes long-horizon planning, and proves particularly effective in challenging settings.

## 5. Conclusion

We study multi-step theorem prediction from a training-free perspective and identified *structural drift* as a core limitation of vanilla in-context learning. To address this issue, we proposed **Pri-TPG**, a training-free framework that leverages retrieved *Theorem Precedence Graphs*, a non-parametric structural prior to capture temporal theorem dependencies and constrain inference. Combined with retrieval-guided graph construction and a stepwise symbolic executor, our framework enables LLMs to function as structured planners

without parameter updates. Experiments on FormalGeo7k benchmark show that our approach substantially outperforms ICL baselines and achieves performance comparable to state-of-the-art supervised methods. More broadly, these results highlight the importance of explicit topological priors for scalable symbolic reasoning, offering a promising paradigm for general large and structured domains.

**Limitations.** Despite its effectiveness, the proposed framework has several limitations. First, it depends on LLM reasoning quality and inference efficiency; repeated calls during multi-step planning dominate computation, limiting efficiency and scalability. This constraint is expected to ease as LLMs continue to improve in both reasoning quality and inference speed. Second, performance on the Hard (L5–L6) tiers remains a frontier for improvement. These problems demand long-horizon global consistency, where a single error can invalidate the entire proof. Although the proposed TPG provides structural guidance, it primarily encodes local precedence rather than global reasoning depth. Developing non-parametric mechanisms to incorporate reasoning depth is an important direction for future work.

## Acknowledgments

This work was jointly supported by the National Natural Science Foundation of China (62437001) and the Fundamental Research Funds for the Central Universities (2253500001, 2243100004).

## Impact Statement

This work studies education-oriented formal geometry solving by combining LLM-based planning with symbolic verification and explicit structural priors in a training-free framework. Its potential benefit is to support tutoring and feedback systems that provide more reliable, checkable solution traces and reduce the cost of maintaining solvers as theorem libraries evolve. We will open-source the code to facilitate reproducibility and follow-up research.

Potential negative impacts include academic misconduct and over-reliance on automated reasoning, which may reduce genuine practice and learning. Since the system cannot guarantee solving every problem, its outputs should not be treated as authoritative. We recommend restricting full-solution outputs in assessment settings and emphasizing step-level guidance over answer delivery in educational use.

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

# Appendix

# A. Implementation Details

### A.1. Retrieval Engineering Pipeline

While the main methodology describes the retrieval-augmented framework, we provide specific details regarding the construction of the candidate theorem pool $\mathcal{L}_q$. The pipeline consists of five stages:

1. **Multimodal Embedding.** For each problem in the training set and the query set, we construct a representation by concatenating the problem text, formal state and the diagram image. This composite is encoded using `jina-embeddings-v4-base-en` to produce a unified dense vector.

2. **Similarity Search.** We perform a brute-force $k$-Nearest Neighbor (kNN) search using cosine similarity to identify the top-$K$ most similar problems from the training set.

3. **Theorem Extraction.** From the retrieved $K$ neighbors, we extract the ground-truth theorem sequences used in their solutions.

4. **Candidate Aggregation.** We aggregate all unique theorems appearing in the retrieved solutions to form the initial query-specific candidate pool.

5. **Ranking and Filtering.** The aggregated candidates are ranked by their frequency of occurrence in the top-$K$ neighbors. We retain the top $N$ frequent theorems to form the final candidate set $\mathcal{L}_q$, which is then dynamically filtered by the solver at each step based on precondition satisfaction.

### A.2. Hyperparameters

Table 7 lists the default hyperparameters used in our experiments on the FormalGeo7K benchmark.

*Table 7.* Inference hyperparameters.

| Hyperparameter | Value |
|---|---|
| *Retrieval Module* | |
| Embedding Model | jina-embeddings-v4-base-en |
| Retrieval Size ($K$) | 200 |
| *Solver Constraints* | |
| Max Steps ($T_{\max}$) | 20 |
| Time Limit per Problem ($\Delta$) | 600s |
| Max Recovery Attempts ($r$) | 3 |
| *LLM Generation* | |
| Temperature | 0.1 |

### A.3. Implementation Details of Candidate Prioritization.

The composite scoring function is instantiated with $\alpha = 5$, $\beta = 10$, and $\gamma = 5$. This mechanism does not impose a strict ordering for the LLM to adhere to; rather, it reorders the candidate list to surface more contextually relevant theorems earlier, thereby enabling efficient truncation of the candidate window passed to the LLM while preserving high-quality options.

**Goal Alignment** ($s_{\text{goal}}$). A theorem receives $\alpha = 5$ if its name contains keywords semantically aligned with the problem's goal type—e.g., "angle", "parallel", or "perpendicular" for angular goals; "length", "pythagorean", or "similar" for metric goals; "area" or "ratio" for areal goals.

**Graph-Based Successor Promotion** ($s_{\text{graph}}$). Immediate successors of $a_{t-1}$ in $G_q$ receive a base score of $\beta = 10$, scaled by their normalized edge frequency $w \in [0, 1]$: $\beta \cdot (0.5 + w)$. Second-order successors (reachable via two-hop paths) receive a reduced score of $4$, modulated by the product of intervening edge weights. This biases the selection toward empirically validated transitions while maintaining a simple linear structure.

**History Penalty** ($s_{\text{hist}}$). Each prior application of theorem $v$ accumulates a penalty of $\gamma = 5$. In recovery mode—triggered when $a_{t-1}$ yields no new predicates—the failed theorem incurs an additional penalty of 20, enforcing exploration of alternative paths.

Importantly, this scoring mechanism serves primarily to facilitate *candidate truncation*: by ranking theorems, we can limit the input window to the LLM (e.g., top-$k$ candidates) without discarding high-potential actions. As shown in Table 8, candidate prioritization substantially improves performance across all difficulty levels, with the largest gains on harder instances, confirming that better truncation preserves useful actions under a limited context window.

*Table 8.* Effect of candidate prioritization on multi-step theorem reasoning across difficulty levels.

| Setting | Total | L1 | L2 | L3 | L4 | L5 | L6 |
|---|---|---|---|---|---|---|---|
| Similarity rank | 70.79 | 92.07 | 76.59 | 55.26 | 50.32 | 41.94 | 16.67 |
| Pri-TPG (Ours) | 84.42 | 98.54 | 93.09 | 78.20 | 64.33 | 58.06 | 23.33 |

All coefficients are derived from general properties of theorem dependency structures rather than dataset-specific tuning, ensuring that the LLM retains primary agency in final theorem selection based on its broader contextual reasoning.

## B. Additional Material

### B.1. Extended Results

Due to space and formatting constraints, several tables in the main text report only aggregated metrics or a subset of difficulty levels. For completeness and reproducibility, we provide the full breakdown results (Total and L1–L6) in this appendix.

*Table 9.* Backbone comparison for our method across difficulty levels.

| Backbone | Total | L1 | L2 | L3 | L4 | L5 | L6 |
|---|---|---|---|---|---|---|---|
| DeepSeek v3.2(Liu et al., 2025) | 83.57 | 98.12 | 92.29 | 77.44 | 63.06 | 56.45 | 21.67 |
| GPT-5 mini(OpenAI, 2025a) | 84.42 | 98.54 | 93.09 | 78.20 | 64.33 | 58.06 | 23.33 |
| Claude 4.5 Sonnet(Anthropic, 2025) | 87.07 | 99.16 | 94.95 | 80.45 | 72.61 | 62.90 | 33.33 |
| Gemini 3.0 Pro(DeepMind, 2025) | 88.43 | 98.75 | 96.01 | 86.09 | 73.89 | 64.52 | 31.67 |
| GPT-5.2(OpenAI, 2025b) | 89.29 | 99.16 | 96.28 | 87.92 | 77.07 | 66.13 | 30.00 |

*Table 10.* Effect of non-parametric structural priors on multi-step theorem reasoning across difficulty levels.

| Setting | Iterative | RAG | TPG | Total | L1 | L2 | L3 | L4 | L5 | L6 |
|---|---|---|---|---|---|---|---|---|---|---|
| Vanilla ICL | ✓ | ✗ | ✗ | 26.29 | 52.19 | 23.67 | 7.89 | 5.10 | 0.00 | 0.00 |
| w/o TPG | ✓ | ✓ | ✗ | 72.64 | 95.62 | 76.06 | 73.68 | 50.32 | 33.87 | 11.67 |
| Pri-TPG | ✓ | ✓ | ✓ | 84.42 | 98.54 | 93.09 | 78.20 | 64.33 | 58.06 | 23.33 |

### B.2. Extended Experiments

**Ablations on GPT-5.2** We used GPT-5 mini for our extensive step-by-step ablations primarily due to the prohibitive API cost of evaluating all 10 variants across the massive FormalGeo7K test set (1,400 problems) with GPT-5.2. To address your concern, we reproduced the core ablation components using GPT-5.2. These results show the same trend as GPT-5 mini, confirming that GPT-5 mini is sufficient for module-level ablation validation, while GPT-5.2 reflects the peak performance of the full framework.

**Pri+Traditional Search** To isolate the prior's effect, we additionally tested (Prior + ForwardSearch) and (Prior + Backward-Search). These results show clear gains from retrieval-based priors: overall accuracy improves from 39.71 to 54.43 and from 35.44 to 48.14, respectively. At the same time, classical search mainly benefits from action-space pruning and cannot fully exploit the richer guidance sequences in the Theorem Precedence Graph (TPG). This is why combining TPG priors with strong trajectory-reasoning models (e.g., LLMs) remains important.

*Table 11.* Ablations on GPT-5.2

| Method(GPT-5.2) | Total | L1 | L2 | L3 | L4 | L5 | L6 |
|---|---|---|---|---|---|---|---|
| ICL LLM | 28.07 | 55.11 | 24.73 | 9.40 | 7.01 | 0.00 | 0.00 |
| Query-adaptive prior | 63.43 | 88.52 | 55.85 | 54.89 | 47.13 | 41.94 | 13.33 |
| Pri-TPG (Ours) | 89.29 | 99.16 | 96.28 | 87.92 | 77.07 | 66.13 | 30.00 |

*Table 12.* Comparison for our Prior combines with traditional search method

| Method | Total | L1 | L2 | L3 | L4 | L5 | L6 |
|---|---|---|---|---|---|---|---|
| ForwardSearch(Zhang et al., 2024) | 39.71 | 58.47 | 41.01 | 34.16 | 16.4 | 5.45 | 4.79 |
| BackwardSearch(Zhang et al., 2024) | 35.44 | 66.43 | 34.98 | 11.78 | 6.56 | 6.09 | 1.03 |
| Prior + ForwardSearch | 54.43 | 72.03 | 63.56 | 40.23 | 37.58 | 11.29 | 8.33 |
| Prior + BackwardSearch | 48.14 | 71.61 | 57.71 | 25.94 | 22.29 | 12.90 | 3.33 |
| Pri-TPG (Ours) | 84.42 | 98.54 | 93.09 | 78.20 | 64.33 | 58.06 | 23.33 |

# C. Prompt Interfaces

To ensure reproducibility, we provide the specific prompts used for both the end-to-end baseline (Direct Solve) and our proposed stepwise solver.

## C.1. Baseline: Direct Solve Prompt

For the direct generation baseline, we employ a Chain-of-Thought (CoT) style prompt that instructs the LLM to output a final answer directly.

---

**Direct Solve System Prompt**

You are an expert in plane geometry problem solving. You will be given a geometry problem that includes both a natural language description and a formal language description (including geometric images and problem text). Your task is to analyze the given geometric problem, understand the known conditions and the goal, and provide a step-by-step solution with clear reasoning and accurate calculations.

**Execution Plan:** 1. Understand the Problem: Identify known conditions and goal (brief, 1-2 sentences). 2. Plan the Solution: State which theorem or formula to use (brief, 1-2 sentences). 3. Solve Step-by-Step: Show key calculations (maximum 5 steps, be concise). 4. Verify the Solution: Quick check if necessary (1 sentence, optional). 5. Final Answer: You MUST end with "FINAL ANSWER: [value]".

**IMPORTANT:** Keep your entire response under 300 words. Be concise but clear.

**Answer Format (CRITICAL):** FINAL ANSWER: [value only, no explanations]
- Integer: 55
- Fraction: 31/2
- Square root: `2*sqrt(21)`
- With pi: `7139*pi/72`
- Expression: `96-8*pi`

**Format Rules:** Use `*` for multiplication, `sqrt()` for roots, and `pi` for $\pi$. NO degree symbols, units, or explanations after the answer.

---

**Direct Solve User Prompt Template**

Problem text: {problem_text}
Formal description of problem image: {image_cdl}
Formal description of problem text: {text_cdl}
Formal description of problem goal: {goal_cdl}

---

## C.2. Pri-TPG: Iterative Solver Prompt

Our method uses a constrained interaction loop where the LLM selects a single theorem call at each step.

You should propose your most promising theorems to try for the current geometry problem. DO NOT output any object names or argument bindings. **Critical:** Candidate theorems are ranked by similarity and executability (readiness scores), but this does NOT mean you should pick the first one. Analyze the goal, current state, and choose theorems that best advance toward solving the problem.

**Rules:**

- Only use theorem names from `candidate_theorems`.

- Do NOT provide geometry objects or arguments; the solver enumerates valid instances automatically.

- Check `theorem_dependencies` (ready vs. blocked) and follow `planning_suggestions`.

- Base your decision on `problem`, `state`, `history`, and `delta_from_last_step`.

- In recovery mode, avoid calls listed in `recent_failure`.

Return STRICT JSON only: {`"calls":` [`"theorem"`]}. No commentary.

---

**Iterative User Prompt Body**

**[INPUT]**
problem: ... (id, statement, formal CDL, goal)
state: ... (summary, delta)

**[CANDIDATES]**
- top_M:
- call: `<theorem>` — status: `<ready/blocked>` — score: `<score>`

...
**[HISTORY]**
- recent steps: ...

**[FAILURE]** (conditional)
- recent failed: ...

**[OUTPUT]**
Exactly one theorem call from candidates, including its branch index.

