# OpenReview forum: "Non-Parametric Structural Priors for Geometry Theorem Prediction"
_ICML.cc/2026/Conference — ICML 2026 regular_

### Official Review · Reviewer_Q4s7 · 2026-03-06

**Soundness:** 3
**Presentation:** 3
**Significance:** 3
**Originality:** 3
**Overall Recommendation:** 4
**Confidence:** 3

**Summary:**

This paper addresses multi-step theorem prediction in automated geometry reasoning. The authors identify "Structural Drift," a phenomenon where vanilla in-context learning performance degrades sharply as reasoning depth increases. To mitigate this, they propose Pri-TPG, a training-free framework that constructs Theorem Precedence Graphs from historical solution traces via multimodal retrieval, and uses them as explicit topological priors to constrain the LLM's theorem selection at each step. Combined with a stepwise symbolic executor and a candidate prioritization scoring function, the method achieves 89.29% accuracy on FormalGeo7K with GPT-5.2, outperforming ICL baselines and matching the best supervised solver. Experiments on Geometry3K and GeoQA further demonstrate competitiveness.

**Compliance With Llm Reviewing Policy:**

Affirmed.

**Final Justification:**

I have no further concerns.

**Key Questions For Authors:**

- How would Pri-TPG be instantiated in a formal reasoning domain where structured solution traces with explicit theorem dependencies are not available, such as general-purpose theorem proving in Lean or Isabelle?
- What is the average number of LLM API calls per problem across difficulty levels, and how does the total inference cost compare to the training cost of supervised baselines like FGeo-HyperGNet?
- Have you investigated learning the prioritization coefficients (alpha, beta, gamma) or replacing the keyword-based goal alignment with an embedding-based similarity, and if so, what was the effect?
- Given that the retrieval pool and TPG are constructed from the training split of the same dataset, how sensitive is the method to distribution shift between training and test problems, particularly for problems involving theorem combinations unseen during training?

**Limitations:**

yes

**Strengths And Weaknesses:**

### Strengths

- The concept of "Structural Drift" is clearly defined and empirically validated with a compelling performance-depth curve (Figure 1), providing a concrete motivation for the proposed approach.
- The framework is entirely training-free at the theorem prediction stage, which is a practical advantage over supervised neural-symbolic solvers that require costly retraining when theorem libraries change.
- The ablation studies are thorough and well-structured, progressively demonstrating the contribution of each component (iterative feedback, RAG, TPG, state-aware prior) with consistent improvement at each stage.
- The method generalizes across multiple LLM backbones (Table 5), supporting the claim that it functions as a model-agnostic reasoning scaffold rather than a model-specific optimization.

### Weaknesses

- The entire evaluation is restricted to plane geometry, and the TPG construction fundamentally depends on the availability of structured solution traces with explicit theorem dependency annotations, making it unclear whether the approach can transfer to other formal reasoning domains (e.g., algebra, combinatorics, or general Lean/Isabelle theorem proving) where such structured traces are not readily available.
- The performance on the hardest problems (L6) drops to 30.00%, significantly below the supervised baseline FGeo-HyperGNet (56.67%), and the authors acknowledge that the TPG only captures local precedence rather than global reasoning depth, which limits the method's applicability to problems requiring long-horizon planning.
- The candidate prioritization scoring function (Eq. 5) relies on manually designed coefficients (alpha=5, beta=10, gamma=5) and keyword-based goal alignment (matching theorem names to goal types like "angle" or "length"), which is a brittle heuristic that may not generalize to theorem libraries with different naming conventions.
- The framework requires repeated LLM API calls at every reasoning step (up to 20 steps per problem with a 600-second timeout), and the paper does not report inference cost, latency, or the total number of API calls, making it difficult to assess practical scalability compared to single-pass supervised solvers.

---

> ### Author Rebuttal · Authors · 2026-03-30
>
> We thank the reviewer for the careful evaluation and constructive questions.
>
> **1. API Calls, Scalability, and Inference Latency.**
> We appreciate the reviewer’s focus on practical scalability. To ensure a fair comparison, all methods were evaluated under a **unified 600s timeout budget** per problem. The average LLM API calls (reasoning steps) across difficulty levels are summarized below.
>
> On L5 and L6, our API calls decrease due to the **600s timeout constraint**. Here, complex theorem instantiations and longer symbolic verification yield fewer total executable steps within the budget.
>
> An immediate optimization is **Multi-step Theorem Chunking**. Using the TPG's structural clusters, the model can predict highly correlated theorem sequences (e.g., a 'tactic block') in one API call rather than iterating individually. This 'coarse-to-fine' strategy significantly reduces inference latency and API overhead.
>
> While supervised baselines (_FGeo-HyperGNet_) have lower inference costs, they require expensive domain-specific training. Our **training-free (ICL)** paradigm eliminates these fixed costs, offering an agile, cost-effective solution for evolving libraries where frequent retraining is prohibitive.
>
> | Method | Total | L1 | L2 | L3 | L4 | L5 | L6 |
> | :--- | :--- | :--- | :--- | :--- | :--- | :--- | :--- |
> | GT | 4.33 | 1.59 | 3.48 | 5.42 | 7.45 | 9.41 | 13.22 |
> | Pri-TPG(GPT-5.2) | 5.71 | 3.02 | 6.14 | 7.57 | 8.41 | 8.25 | 6.60 |
>
>
> **2. Performance on Hardest Problems (L6) and Long-horizon Planning.**
> We thank the reviewer for this observation. While our L6 accuracy (30.00%) trails the fully supervised baseline (56.67%), we acknowledge that the TPG primarily captures local precedence, and L6 performance is severely constrained by the **600s timeout limit** and increased symbolic verification depth. Addressing long-horizon planning, potentially via **multi-step theorem chunking**, is a pivotal next step in our roadmap. We contend these results provide a robust **proof-of-concept** for scaling structured reasoning without domain-specific fine-tuning.
>
> **3. Robustness of Candidate Prioritization Scoring (Coefficients and Heuristics).**
> We thank the reviewer for the critical reflection on our prioritization function. We respectfully argue these design choices are not "brittle heuristics," but are strategically grounded in formal reasoning requirements.
>
> (1) **Embedding-based similarity:** We deliberately chose symbolic keyword matching over embeddings to ensure **logical grounding**. In formal geometry, the distinction between "angle" and "area" is absolute. Embedding models often suffer from **semantic hallucination**, conflating logically distinct concepts appearing in similar contexts.
>
> (2) **Parameter Tuning:** While learning these parameters could optimize performance, we prioritized a transparent, heuristic-driven approach to maintain a **strictly training-free (ICL) paradigm**.
>
> (3) **Naming Conventions:** While names vary, the **structural dependencies** are topology, not just naming. Adapting to new libraries requires only a one-time mapping of goal types. This maintains our **Training-free (ICL)** integrity, avoiding expensive retraining when libraries evolve.
>
> **4. Sensitivity to Distribution Shift and Unseen Combinations.**
> We thank the reviewer for this insightful concern. Our method is inherently robust to distribution shifts because the **Theorem Precedence Graph (TPG)** provides **atomic logical priors** rather than monolithic procedural templates. Unlike simple path retrieval, the TPG decomposes historical traces into a directed graph of **pairwise dependencies** (e.g., _Theorem A typically precedes Theorem B_). For novel test problems, the TPG allows the LLM to **recombine these local logical building blocks** to navigate new search spaces. Sustained performance gains on complex benchmarks (L5/L6), involving long-chain reasoning paths rarely seen during training, demonstrate that TPG acts as a **structural constraint** facilitating **compositional generalization** over trajectory memorization.
>
> **5. Transferability to Other Formal Domains (e.g., Lean/Isabelle).**
> TPG construction relies on structured traces. In domains where such annotations are unavailable, one possible path is abstracting tactic scripts into macro-level dependencies via lightweight syntax analysis or trace parsing. Developing reliable extraction of local precedence from unstructured or semi-structured tactic logs remains our key future work.

---

> > ### Author Rebuttal · Reviewer_Q4s7 · 2026-03-31
> >
> > I have no further questions and will keep my positive score.

---

> > > ### Author Response · Authors · 2026-04-05
> > >
> > > Thank you for your careful evaluation and valuable feedback. We are glad that all concerns have been resolved and appreciate the reviewer maintaining the positive evaluation.

---

### Official Review · Reviewer_x9wj · 2026-03-10

**Soundness:** 3
**Presentation:** 3
**Significance:** 3
**Originality:** 3
**Overall Recommendation:** 4
**Confidence:** 3

**Summary:**

This paper studies training-free multi-step theorem prediction for formal geometry problem solving. It proposes Pri-TPG, a neural-symbolic framework that uses Theorem Precedence Graphs as non-parametric structural priors derived from retrieved historical solution traces. The method combines retrieval-based graph construction, stepwise LLM theorem selection, and symbolic execution with state-aware pruning and candidate prioritization. The empirical evaluation is conducted primarily on FormalGeo7K, with additional results on Geometry3K and GeoQA. The paper reports that Pri-TPG substantially outperforms vanilla ICL baselines and achieves performance competitive with strong supervised neural-symbolic methods. The main contributions are the formulation of theorem precedence graphs for multi-step theorem prediction, a training-free inference framework integrating retrieval and symbolic execution, and an empirical study on formal geometry benchmarks.

**Compliance With Llm Reviewing Policy:**

Affirmed.

**Final Justification:**

The authors have adequately addressed my main concerns regarding soundness and scope, and I have therefore upgraded my overall recommendation. In particular, the rebuttal strengthens the empirical case by adding GPT-5.2 results, clarifies the retrieval protocol and hold-out separation, and better distinguishes theorem prediction under provided formal states from broader end-to-end geometry problem solving. I now view the work as a useful and reasonably strong contribution on training-free neural-symbolic theorem prediction, with strong results on FormalGeo7K.

**Key Questions For Authors:**

1. The strongest result is reported with Pri-TPG (GPT-5.2), but the main ablations are conducted only with GPT-5 mini. Could the authors provide the key controlled ablations with GPT-5.2 as well (e.g., Vanilla ICL, w/o TPG, and other main variants under the same step-wise execution setting)?
2.Could the authors describe the retrieval database and test-time retrieval protocol more precisely? What measures are taken to avoid retrieval overlap or leakage from highly similar solution traces?
3.In Table 2, some baselines are unreported, and some methods are evaluated only on subsets that can be parsed into the required formal language. Could the authors clarify how comparable these numbers really are, and whether the “best reported” / “highly competitive” claims should be interpreted with any caveats?
4.The paper’s title and initial description emphasize multi-step theorem prediction, but the experiments and analysis focus on geometry problem solving, specifically formalizing and solving geometry theorems. Could the authors clarify the distinction between these two tasks in the context of their approach? Is the framework designed solely for theorem prediction, or is it also applicable to the broader geometry problem-solving process?

**Limitations:**

yes

**Strengths And Weaknesses:**

Soundness.
The empirical validation has several important gaps. The evaluation is restricted to theorem prediction from provided formal states rather than end-to-end geometry solving, so the paper does not establish the broader capability it sometimes suggests. The strongest result is reported with GPT-5.2, but the key ablations are conducted only with GPT-5 mini, which leaves the effect of the proposed structural prior insufficiently isolated. Table 2 contains missing baseline results, and some comparisons are reported only on subsets that can be parsed into the required formal language, which reduces comparability. The retrieval setup is also not specified clearly enough to rule out concerns about overlap or retrieval leakage.

Presentation.
The paper is readable, but the framing is broader than the evidence supports. It repeatedly moves from theorem prediction under provided formal states to broader claims about geometry problem solving and structured reasoning more generally. The distinction between what is directly demonstrated and what is inferred is not always drawn sharply enough, especially in the discussion of scalability, adaptability, and generality.

Significance.
The problem is relevant, but the contribution is narrower than the paper implies. This is a theorem-prediction method for formal geometry under a controlled evaluation setup, not a broader advance in theorem proving. The results on FormalGeo7K are strong, but the practical scope is limited by the reliance on provided formal states, historical solution traces, and a retrieval-based prior whose robustness is not fully characterized.

Originality.
The main contribution is the use of retrieved Theorem Precedence Graphs as structural priors for theorem prediction. This is a reasonable and useful design idea, and it clearly distinguishes the method from vanilla ICL. That said, the novelty is primarily at the level of system integration rather than a new reasoning model or learning principle. The paper should be understood as a targeted neural-symbolic design contribution rather than a broader methodological advance.

---

> ### Author Rebuttal · Authors · 2026-03-30
>
> We thank the reviewer for the detailed assessment and constructive feedback.
>
> **1. Ablations with GPT-5.2.**
> We used GPT-5 mini for our extensive step-by-step ablations primarily due to the prohibitive API cost of evaluating all 10 variants across the massive FormalGeo7K test set (1,400 problems) with GPT-5.2. To address your concern, we reproduced the core ablation components using GPT-5.2. These results show the same trend as GPT-5 mini, confirming that GPT-5 mini is sufficient for module-level ablation validation, while GPT-5.2 reflects the peak performance of the full framework.
>
> | Method (GPT-5.2) | Total | L1 | L2 | L3 | L4 | L5 | L6 |
> | :--- | :--- | :--- | :--- | :--- | :--- | :--- | :--- |
> | ICL LLM | 28.07 | 55.11 | 24.73 | 9.40 | 7.01 | 0.00 | 0.00 |
> | Query-adaptive prior | 63.43 | 88.52 | 55.85 | 54.89 | 47.13 | 41.94 | 13.33 |
> | Pri-TPG (Ours) | 89.29 | 99.16 | 96.28 | 87.92 | 77.07 | 66.13 | 30.00 |
>
>
> **2. Retrieval Protocol and Leakage.**
> There is no retrieval leakage. The retrieval database is built _exclusively_ from the training split, and all test-time retrieval follows strict hold-out separation. The detailed construction and protocol are documented in **Appendix A**; in the final version, we will also highlight the key points (database scope, retrieval pipeline, separation guarantee) in the main text for immediate clarity.
>
> **3. Table 2 Baseline Fairness and Data Subsets.**
> For Geo3K, we adopted the established subset following our primary competitive baseline, _HyperGNet_. Although this subset removes duplicate-diagram problems, it still covers over 92% of the dataset. This choice ensures direct and fair alignment with the strongest comparator. Missing baselines are absent because they are closed-source or structurally incompatible with HyperGNet's parsing protocol. We will make these choices explicitly in the final paper.
>
> **4. Theorem Prediction vs. Geometry Problem Solving.**
> We thank the reviewer for this opportunity to clarify the task. While **Geometry Problem Solving (GPS)** encompasses a pipeline including **Auto-formalization** and **Theorem Prediction**, each represents a distinct research challenge with its own specialized literature. In alignment with established benchmarks (e.g., _HyperGNet_), our work follows the standard paradigm of assuming predefined formal states (as noted in Line 134) to isolate and rigorously address the **multi-step theorem prediction bottleneck**.
>
> We chose geometry not as a narrow application, but as a **high-fidelity testbed** for our **Theorem Precedence Graph (TPG)**. The dense, well-defined logical dependencies in geometry theorems provide an ideal environment to demonstrate how non-parametric structural priors can effectively prune vast search spaces. Furthermore, the core philosophy of our approach, leveraging historical solution traces to construct structural priors, is **domain-agnostic** and conceptually applicable to any multi-step theorem prediction task, whether in formal logic or theorem proving. We will refine the Introduction to more explicitly position theorem prediction as the strategic core of GPS and clarify its role within the broader solving pipeline.
>
> **5. Originality and Contribution Scope.**
> We thank the reviewer for the characterization of our work as a "reasonable and useful design." However, we respectfully argue that our contribution extends beyond "system integration" into a **new paradigm for training-free symbolic reasoning**.
>
> (1) **Shift from Parametric to Structural Priors:** Traditionally, "methodological advances" in neural-symbolic reasoning have relied on gradient-based fine-tuning to bake logic into model weights. Our work introduces a **broader learning principle**: that LLMs can achieve SOTA reasoning performance by leveraging **externalized, non-parametric structural priors** (TPG) derived from historical traces. This demonstrates that "reasoning competence" can be decoupled from "model parameters," a significant conceptual shift for the ICL community.
>
> (2) **A Targeted Solution to "Structural Drift":** We identified and formalized **Structural Drift** as a fundamental failure mode of LLMs in long-chain reasoning. The TPG is not merely an integrated component; it is a **mathematically-grounded constraint mechanism** that prunes the search space based on topological precedence. This provides a reproducible methodology for grounding LLMs in any domain with structured solution histories.
>
> (3) **Beyond Geometry:** While we chose geometry as our testbed, the principle of **Retrieval-Augmented Structural Guidance** is a general-purpose advance. It offers a scalable alternative to supervised models, especially in evolving environments where retraining is prohibitive.

---

> > ### Author Rebuttal · Reviewer_x9wj · 2026-04-03
> >
> > My main concerns have been sufficiently addressed, and I am therefore increasing my score.

---

> > > ### Author Response · Authors · 2026-04-05
> > >
> > > Thank you for your follow-up and for raising the score. We truly appreciate your time, consideration, and thoughtful reading of our rebuttal. We are very grateful that our response helped clarify your main concerns.

---

### Official Review · Reviewer_gXFB · 2026-03-14

**Soundness:** 3
**Presentation:** 3
**Significance:** 3
**Originality:** 3
**Overall Recommendation:** 5
**Confidence:** 4

**Summary:**

The paper proposes Pri-TPG, a neuro-symbolic inference-time method for solving geometry problems. First, it extracts historic solution traces into a symbolic graph form, where an edge from theorem A to theorem B indicates theorem A must be applied before B. Then, it uses this prior to decide which theorems to apply at a given point when solving the current problem. Pri-TPG shows strong results on FormalGeo7k.

**Compliance With Llm Reviewing Policy:**

Affirmed.

**Final Justification:**

The rebuttal has reinforced my prior assessment, and I believe this paper should be accepted due to the simple ideas, detailed ablations, and effective results.

**Key Questions For Authors:**

- How do you see the methods in this paper extending to other domains such as general theorem proving in Rocq or Lean? While I understand if this is not possible due to the time constraint, it would be great if the authors could show any preliminary experiments highlighting the success of their approach on other domains.
- How does Claude Code or Codex with agent mode perform on this task? Can any of the ideas in the paper apply on top of these agent frameworks that already have access to an execution engine?

**Limitations:**

yes

**Strengths And Weaknesses:**

Soundness
- Compared to vanilla ICL, the proposed methods shows very strong improvements, particularly using GPT-5 mini. It is clear that for harder problems with longer reasoning chains, Pri-TPG outperforms ICL baselines.
- The ablations provide good coverage and understanding for the utility of the different components. I wish harder problems could have been used in Table 5 to better understand the impact of the underlying base model on the results.
- Overall, I would have liked to see more validation of the techniques on more domains or harder problems.

Presentation
- Overall, the work is easy to follow.

Significance
- One issue limiting the generality of the approach is that it relies on the domain having a relatively constrained action space. The problems in the dataset use up to 10 theorems, but real-world proving can use hundreds or more theorems. It is unclear to me whether these techniques can generalize beyond the simple domain considered in the paper.
- One baseline that was not compared to is the latest wave of Codex/Claude Code models with agent-mode and execution-mode allowed. This way, the model would be allowed to run the symbolic executor by itself, and give self-feedback.

Originality
- The ideas are simple, but novel, original, and effective.
- The ideas of retrieval on the graph-level and leveraging the symbolic graph hierarchy have been applied in older knowledge graph/logical reasoning literature, they are relatively rare in the theorem proving.
- I can imagine combining the ideas in this paper with some of the neuro-symbolic solvers that are training based for an even stronger system. Do the authors believe that training can boost the performance of any aspect of this approach?

---

> ### Author Rebuttal · Authors · 2026-03-30
>
> We thank the reviewer for the positive evaluation and constructive suggestions.
>
> **1. Table 5 on Harder Problems.**
> We agree that harder subsets provide especially valuable insight. The complete L1-L6 breakdown is already reported in **Appendix B.1 (Extended Results)**. Due to strict main-text space limits in the submission, we could not include the full per-level table there; following your suggestion, we will prioritize surfacing the harder-subset (high-difficulty) results more explicitly in the main text of the final version, as these are likely of greatest interest to readers.
>
> **2. Constrained Action Space and Scalability.**
> We appreciate the reviewer's perspective. Our **current primary goal** is to establish the fundamental efficacy of non-parametric structural priors within the **Training-free (ICL) paradigm**. The substantial performance boost achieved on the **FormalGeo7k** benchmark (89.29% accuracy) successfully validates our core hypothesis: that explicit topological constraints can effectively mitigate "Structural Drift" even in challenging geometry reasoning tasks.
>
> While the current dataset represents a controlled environment, the **TPG mechanism** is specifically engineered to counteract the **combinatorial explosion** inherent in larger action spaces. As the number of theorems scales, we expect the utility of TPG to transition from a "helpful guide" to a **"strict necessity"** for maintaining proof coherence in increasingly sparse search spaces.
>
> Extending this framework to broader domains with hundreds of theorems, and integrating it with **autonomous agents** for large-scale theorem proving, represents a **pivotal next-step direction** in our research roadmap. We contend that the current results provide a robust **proof-of-concept** that structural priors are a scalable alternative to supervised fine-tuning, laying the necessary foundation for these future explorations.
>
> **3. Synergy with Trained Models.**
> We agree. Our retrieval-based prior is orthogonal to training and can be integrated as a plug-and-play search module for trained neuro-symbolic solvers, potentially improving their current ceilings. We will make this compatibility clearer in the final manuscript.
>
> **4. Extending to other domains (e.g., Lean, Rocq).**
> Thanks for insightful question. We agree that the transition from geometry to general-purpose theorem proving (e.g., Lean, Rocq) is the ultimate frontier for automated reasoning. While a rigorous empirical evaluation in Lean/Rocq requires a dedicated symbolic execution interface beyond the rebuttal timeline, we argue that our approach is conceptually **domain-agnostic** and holds significant potential for these domains.
>
> Following the neural-symbolic synergy validated by SOTA models like **AlphaGeometry**, our work translates this principle into a **training-free (ICL) paradigm**. Specifically, in tactic-based languages, tactic applications and proof traces can be mapped as nodes and directed edges; the TPG then acts as a **topological filter** to prune the search space, a mechanism arguably more critical in the vast and sparse environments of Lean than in geometry. Furthermore, this **non-parametric framework** natively accommodates **evolving theorem libraries** without retraining, offering a scalable advantage that can complement supervised learning as the reviewer anticipated.
>
> **5. Agent-based Frameworks (Codex / Claude Code).**
> Thanks for this forward-looking suggestion. Pri-TPG could indeed serve as a structural planner on top of agent-style execution loops. However, a fair empirical comparison with agent-mode systems requires substantial additional integration (tool orchestration, controlled execution policies, and budget-matched evaluation), which is difficult to complete reliably within this revision period. We therefore list this as explicit future work and avoid over-claiming without controlled experiments.

---

> > ### Author Rebuttal · Reviewer_gXFB · 2026-04-03
> >
> > I thank the authors for their rebuttal, and believe my score adequately reflects the contributions of the paper.

---

> > > ### Author Response · Authors · 2026-04-05
> > >
> > > Thank you for your valuable suggestions and active engagement. We are glad that our rebuttal has adequately addressed the concerns, and we sincerely appreciate your positive assessment and support for our work.

---

### Official Review · Reviewer_yMsR · 2026-03-15

**Soundness:** 3
**Presentation:** 3
**Significance:** 2
**Originality:** 3
**Overall Recommendation:** 4
**Confidence:** 4

**Summary:**

This paper introduces Pri-TPG, a method for guiding a formal geometry theorem prover using priors over the topological order of theorem applications in proofs. Rather than relying on training, the paper proposes a non-parametric prior that is derived from empirical statistics computed over retrieving similar theorems and proofs from a database. Then, at each step in the proof, the LLM can only select from a smaller set of candidate theorems from the background library, with this set being derived from the retrieved proofs and the current proof state. Experiments on 3 benchmarks of formal geometric problem solving show clear gains over several LLM-based baselines including inference-only methods and others requiring fine-tuning.

**Compliance With Llm Reviewing Policy:**

Affirmed.

**Final Justification:**

Overall I think this is a solid contribution to geometry problem solving: the method is sound and the results are good. The authors clarified several of my concerns, and agreed to make the scope of the paper more clear upfront (it might be difficult to apply to more general theorem proving settings as is, so that should be clear in the title and abstract which only implicitly indicate this when the abstract mentions the benchmark the method was evaluated on). Trusting the authors will address this minor framing issue, I recommend acceptance.

**Key Questions For Authors:**

1. Does the environment enumerate all possible ways to apply a given theorem from the library in a given state? How large is the branching factor after choosing a particular theorem?
2. Have the authors tried a pure search-based baseline combined with the Pri-TPG prior (or not, just given the whole library)? It would be informative to know how strong is the prior and how much need is there still for the LLM after the action space has been drastically restricted.

**Limitations:**

I believe the authors should be more explicit that it is difficult to apply the method as is to broader formal theorem proving tasks.

**Strengths And Weaknesses:**

The paper presents a sound idea of using the structure of retrieved proofs for similar theorems to guide a theorem prover. It's a simple principle that I had not seen yet, and I found the idea compelling and well-motivated. The paper is well written and the presentation is overall polished.

The experimental evaluation shows clear gains on top of a range of relevant benchmarks and ablations. While reading the paper I anticipated several ablations I would like to see (e.g., RAG and just using a global prior), and found them to be included later in the evaluation, with clear results. The method scales well with large retrieved sets (the authors tried up to 200 retrieved proofs), whereas RAG tends to plateau much earlier.

For significance, the paper uses a broad motivation in automated reasoning and theorem proving (and the title and abstract reflect this). However, the method seems difficult to apply to other general theorem proving settings (e.g., theorem proving in Lean or Rocq), since the action space there is not really just a set of theorems (or even tactics), as there is no way to enumerate all the ways to apply a given theorem/tactic in a given proof state (there are often infinite), and proofs can branch (and branches are not even independent, since they can share metavariables). Thus, I believe it is better to reflect the narrower scope early in the paper. Focusing on geometric problem solving is not by itself an issue, but since the method as is is not readily applicable to other tasks, this should be clear very early.

Minor: in the Evaluation section, "Following Zhang et al. (Zhang et al 2025)", by convention the authors should use \citet instead of \citep to avoid repeating Zhang et al.

---

> ### Author Rebuttal · Authors · 2026-03-30
>
> We thank the reviewer for the thoughtful evaluation and constructive suggestions.
>
> **1. Scope and Limitations.**
> We agree that the current method is tailored to geometry problem solving. While retrieval-based prior guidance may generalize in principle, adapting it to broader formal environments (e.g., Lean, Rocq) introduces a dedicated symbolic execution interface. We will make this scope boundary explicit early in the final version.
>
> **2. Branching Factor and Environment Enumeration.**
> Yes, the environment enumerates all valid ways to apply a selected theorem, and the resulting branching factor is typically from a few to several dozens. Following prior work (e.g., _Inter-GPS_, _FGeo-HyperGNet_), we encapsulate this low-level object enumeration in the symbolic solver engine, outside our main modeling scope, and we compare with the baselines under the same time budget in Table 1. A promising next step is to predict theorem _objects_ directly to bypass this enumeration. Your observation pinpoints a real bottleneck in current symbolic execution pipelines.
>
> **3. Search Baseline with Prior.**
> Table 1 already includes pure search baselines (ForwardSearch, BackwardSearch). To isolate the prior's effect, we additionally tested (Prior + ForwardSearch) and (Prior + BackwardSearch). These results show clear gains from retrieval-based priors: overall accuracy improves from **39.71** to **54.43** and from **35.44** to **48.14**, respectively. At the same time, classical search mainly benefits from action-space pruning and cannot fully exploit the richer guidance sequences in the Theorem Precedence Graph (TPG). This is why combining TPG priors with strong trajectory-reasoning models (e.g., LLMs) remains important.
>
> | Method | Total | L1 | L2 | L3 | L4 | L5 | L6 |
> | :--- | :--- | :--- | :--- | :--- | :--- | :--- | :--- |
> | ForwardSearch | 39.71 | 58.47 | 41.01 | 34.16 | 16.4 | 5.45 | 4.79 |
> | BackwardSearch | 35.44 | 66.43 | 34.98 | 11.78 | 6.56 | 6.09 | 1.03 |
> | Prior + ForwardSearch | 54.43 | 72.03 | 63.56 | 40.23 | 37.58 | 11.29 | 8.33 |
> | Prior + BackwardSearch | 48.14 | 71.61 | 57.71 | 25.94 | 22.29 | 12.90 | 3.33 |
>
> **4. Minor.**
> We will fix \citep to \citet.

---

> > ### Author Rebuttal · Reviewer_yMsR · 2026-04-03
> >
> > Thank you for the response, and I appreciate the new results adding the prior to the search-only baselines. I maintain my positive assessment of the paper.

---

> > > ### Author Response · Authors · 2026-04-05
> > >
> > > Thank you for your careful review and constructive feedback. We are glad that our rebuttal resolved the concerns, and we sincerely appreciate your support.

---

### Decision · Program_Chairs · 2026-04-30

**Decision:**

Accept (regular)

**Comment:**

This paper introduces Pri-TPG, a training-free framework for multi-step theorem prediction in formal geometry problem solving. The core contributions are: (1) the identification and empirical characterization of Structural Drift (the sharp performance degradation of vanilla ICL as reasoning depth increases), (2) Theorem Precedence Graphs (TPGs), directed graphs encoding temporal dependencies from historical solution traces, and (3) a retrieval-augmented, stepwise inference pipeline that uses TPGs as topological constraints to prune the theorem search space without any gradient-based training.

Reviewers consistently praised the clarity of the Structural Drift motivation, the thoroughness of the ablation studies, and the model-agnostic nature of the framework. The rebuttal further strengthened the empirical case by providing GPT-5.2 ablations that confirm trends observed with GPT-5 mini, and by clarifying the retrieval protocol and hold-out separation to rule out leakage concerns.

Several limitations were raised and (partially) addressed. All reviewers noted that the method is tailored to formal geometry and relies on structured solution traces with explicit theorem dependency annotations. Extension to general-purpose theorem proving environments is non-trivial. The paper's title and abstract should make this scope boundary explicit early on. On the hardest problems, Pri-TPG is outperformed by the supervised baseline FGeo-HyperGNet. The authors attribute this partly to the 600-second timeout and acknowledge that the TPG captures local precedence rather than global long-horizon structure. This gap should be discussed in depth in the final version. The candidate scoring function uses manually tuned coefficients and keyword matching. While the authors defend these choices as appropriate for formal reasoning, the sensitivity of results to these design decisions warrants more explicit discussion.

The metareviewer agrees with the reviewers that this paper is a solid contribution to training-free neural-symbolic reasoning for geometry problem solving. The paper is likely to be built upon by others working on structured LLM reasoning.